# Heartfulness in Vegans, Vegetarians, and Omnivores

**DOI:** 10.3390/ijerph20064943

**Published:** 2023-03-11

**Authors:** Antonia Voll, Leonardo Jost, Petra Jansen

**Affiliations:** Faculty of Human Sciences, University of Regensburg, Universitätsstraße 31, 93053 Regensburg, Germany

**Keywords:** mindfulness, heartfulness, diet, compassion, sustainability

## Abstract

Background: The primary goal of this study was to investigate the relation between the choice of a vegan or vegetarian diet as a criterion of sustainability and the aspect of heartfulness. We also analyzed which demographic, diet-related, and mindfulness practice-related variables could predict the different facets of heartfulness. Methods: In total, 419 persons participated. After providing demographic, diet-related, and mindfulness practice-related information, participants completed a gratitude questionnaire, a self-compassion scale, a compassion scale, and an equanimity scale. Results: The results show that vegans and vegetarians indicated higher scores than omnivores in some aspects of heartfulness, such as both self-compassion scales. These effects could not be shown for the two equanimity scales and for the gratitude questionnaire. Most aspects of heartfulness could either be predicted by demographic or diet-related variables. The best predictors of the elements of heartfulness were the ecological, ethical, or health-related reasons for choosing their diet stated by the participants, as well as the importance the participants attached to nutrition. Conclusion: This study provides evidence that vegans and vegetarians scored higher in several aspects of heartfulness. Vegans tended to score even higher than vegetarians. Both demographic and diet-related variables could predict heartfulness.

## 1. Introduction

“You are what you eat.” This is a proverb we often hear in everyday life. However, this saying has a certain truth to it. Indeed, there are obvious connections among certain personality traits, values, habits, and eating behavior. The meta-analysis by Holler et al. [1] provides a far-reaching overview of some of these correlations. According to this analysis, omnivores tend to be more authoritarian, socially dominant, bias-oriented, and self-centered than vegetarians and vegans. In contrast, vegetarians and vegans tend to be more open to new experiences, compatible, spiritual, intelligent, and empathetic [2]. The present study investigates whether the persons who can be classified into the three most relevant categories of omnivore, vegetarian, and vegan differ in the aspects of heartfulness.

The definition of omnivores seems to be clear; omnivores include all those individuals who do not place any restrictions on their diet. This means they do not exclusively eat plant or animal foods [3]. In contrast, a universal definition of vegetarian diet is much more difficult to pin down. According to Ruby [4], there are various interpretations of this label, as many self-identifying vegetarians still consume red meat or fish [5]. The most common understanding of a vegetarian diet is “the practice of abstaining from the consumption of meat, animal flesh, or animal slaughter by-product” [6]. In 2022, around 7.90 million people in Germany classified themselves as vegetarians [7]. The generic term vegetarian can be further differentiated into more specific diets [8]. One of these subgroups represents vegan eating behavior. The vegan diet is defined as the “avoidance of all animal products” [9]. This includes meat, fish, milk, eggs, and honey. In Germany, 1.58 million people described themselves as vegan in 2022 [10].

### 1.1. Reasons for Different Dietary Patterns

Research that has focused on the psychological reasons for adhering to a vegetarian or vegan diet has demonstrated that one of the Big Five personality traits, agreeableness, is positively related to adopting a vegan or vegetarian diet [11]. Health and ethical reasons were indicated most often for choosing a vegan diet [12]. Vegetarians were also characterized by a different food-related motivation profile with respect to low, medium, and high meat eaters. For example, they more often wished to prepare their meals themselves, as they wanted to eat everything as purely as possible. They also reported distinctive taste- and animal welfare-related reasons to justify their abstinence from eating meat [13]. However, the reasons given for the different diets are very diverse. As mentioned above, an important reason is a concern for preserving the environment. In his literature review, Rosenfeld concluded that animal welfare, health, and environmental reasons were the most dominant motivators for a vegetarian diet within developed Western nations [14].

According to a study conducted in Germany, the motivation for a vegan diet in particular was also explained by these three factors, with animal-related motives being the most important (89.7%) and environment-related motives being the least important (46.8%) in the sample [15]. A recent study assessed the prevalence of the possible presence of eating disorders in young women following a traditional (omnivorous), vegetarian, or medical diet. It was shown that uncontrolled and emotional eating was the least common in women on a vegetarian diet. Vegetarians mentioned both ethical and health reasons as motivations for choosing their diet [16]. Consequently, it can be assumed that this is a conscious and sensitive group of consumers.

Because this study was conducted in Germany, we also want to elaborate on German vegans, vegetarians, and omnivores in particular. It was shown that vegetarians tend to better meet the dietary reference values of the German Nutrition Society than omnivores [17]. In addition, the meat consumption of male students was on average twice as high as the meat consumption of female students in a study of dietary patterns in German students [18]. Especially in Germany, the vegan and vegetarian diets are on the rise. In a recent study, 75% of vegans and vegetarians identified curiosity as the reason for buying non-meat products, while 71% named animal welfare, and 64% chose the climate as reasons for preferring plant-based food [19]. As the vast majority of Germans (over 93%) are either undenominational or Christian [20] and Christianity does not prohibit any food, we chose to neglect possible religious reasons for the diet choice in this study. 

### 1.2. Dietary Patterns and Sustainability

General ecological concerns are also often cited as reasons for a vegetarian diet [21]. This motivation is justified by Chai et al. [22]. It was evident that participants with a diet rich in plant-based foods showed lower environmental impacts measured with their greenhouse gas emissions, cumulative energy demand, and land occupation [23]. A meta-analysis supports the conclusion that diets higher in plant-based foods and lower in animal-based food are beneficial for environmental sustainability [24].

### 1.3. Mindfulness and Sustainability

Next to dietary patterns, the role of mindfulness in sustainable behavior is discussed regarding its role in choosing a diet [25]. According to Kabat-Zinn, generally, mindfulness is “paying attention in a particular way: on purpose, in the present moment, and nonjudgmentally” [26]. In modern psychology, mindfulness is understood as an approach that can be used to improve awareness [27]. Additionally, mindfulness causes skillful responses to mental processes that can lead to emotional distress and inappropriate behavior [28]. Mindfulness can be differentiated into two main aspects [29]. First, mindfulness solidifies attention and awareness in the experience of the present moment. This experience has different facets, including “body sensations, emotional reactions, mental images, mental talk, and perceptual experiences” [29]. Second, mindfulness also represents an open and accepting attitude toward one’s own experiences. This attitude includes an inviting, open-minded and curious element but also a non-reactive, detached, and passive orientation [29].

Mindfulness can enhance sustainable behavior by increasing awareness of automatic behavior [30]. Experimentally, no direct link between mindfulness and sustainable attitude or behavior has been established yet [31]. Possible intervention studies to investigate whether mindfulness training can foster sustainable consumption behavior are rare [31]. Nevertheless, no evidence of mindfulness practice direct effect on sustainable consumption behavior and attitudes was found [32]. However, in a correlational study, Jansen et al. found an indirect connection between attitudes toward sustainability and the aspects of mindfulness of inner and outer awareness, and insight with the mediator of pro-sociality. In addition, with the mediator of nature connectedness, an indirect relationship between both external awareness and insight, and the attitudes towards sustainability was found [33]. A recent study on how their social media use shapes women’s body images showed that those who practiced a vegetarian diet felt less internalized pressure on their appearance. It is possible that this occurrence is due to vegetarianism being actively promoted online right now [34]. This demonstrates that a more sustainable diet is connected with a more mindful attitude toward the own body. Furthermore, it was shown that the observing effect of mindfulness was correlated with the explicit attitude towards vegetarian food and with goal intention in the framework of sustainable behavior change [35]. The mindfulness aspects mentioned here all have an awareness component, but in 2004, Kabat-Zinn already stated that mindfulness also has a gentle emotional quality that can be described as heartfulness [36].

### 1.4. Heartfulness

According to Voci et al. [37], heartfulness can be divided into two dimensions. First, it includes self-compassion [38], that is, heartfulness toward oneself. Self-compassion is described as “being touched by and open to one’s own suffering, not avoiding or disconnecting from it, generating the desire to alleviate one’s suffering and to heal oneself with kindness” [39]. It also involves understanding one’s pain, failures, and shortcomings without judging them. Self-compassion can be subordinated to self-kindness, mindfulness, and common humanity. Self-kindness consists in showing kindness and understanding towards oneself instead of harsh judgment and self-criticism. Mindfulness in this context is understood as “holding one’s painful thoughts and feelings in balanced awareness rather than over-identifying with them” [39]. Humanity is defined as considering one’s own experiences as part of the more extensive human experience rather than separating and isolating them from it.

On the other hand, heartfulness can also take the form of gratitude [40] directed toward others. It consists of two elements: the recognition that a positive outcome was achieved for oneself and the recognition that an external source, that is, another person, is responsible for it. Lazarus and Lazarus have described gratitude as an “empathic emotion” whose roots lie in the ability to empathize with others [41]. The aspects of heartfulness, gratitude, and self-compassion are highly correlated [37]. Regarding the connection to sustainability, a statistically significant positive correlation was also directly confirmed between gratitude and pro-sociality [42]. Feeling grateful also directly influences sustainability and the selection of sustainable products [43]. 

In addition to the differentiation of Voci et al., the emotional aspect of heartfulness could also be explained by the Buddhist concept of Brahmaviharas, which are the four Buddhist virtues of love, compassion, empathetic joy, and equanimity. Living a lifelong virtuous life from the heart with kindness and equanimity are prime qualities for a person living in the heart [44]. The Brahmaviharas, also called the four immeasurables, can be seen as sustainability virtues [45]. In this study, we focus on the two aspects of compassion and equanimity. According to Singer and Klimecki [46], compassion is defined as the concern for another person’s suffering, which might result in the motivation to help. It was shown that compassion is positively linked to sustainable purchase criteria, and the willingness to pay extra for fair trade clothes can be enhanced with brief compassion training [47]. Equanimity describes a stable and balanced state and is expressed in a balanced reaction to joy as to misery [48].

Various mindfulness meditation techniques are used to achieve a balanced state of mind. Heartfulness meditation in particular is shown to have a very positive impact in different contexts; e.g., it was used to reduce the perceived stress level of people during COVID-19 by cultivating the quality of empathy, acceptance, and individual peace [49]. 

As Niemiec calls heartfulness “the application of character strengths” and “a metaphor for positive action“ [50], it is assumed that this action might express itself in fundamental aspects of life such as education, income, or diet. There is not much literature regarding the concept of heartfulness yet [37], which is why an exploratory approach is needed to determine which general attributes of a person might influence the extent of their heartfulness.

To summarize, heartfulness is an aspect of mindfulness [36], and mindfulness has previously been indirectly connected with sustainability [33]. Presumably, heartfulness as a “positive action” [50] itself is related to sustainable behavior, which might reflect in the choice of a plant-based diet [23]. To our knowledge, no study has investigated this possible connection between heartfulness and the choice of a sustainable plant-based diet yet.

### 1.5. Goal of This Study

The study’s primary goal is to investigate the relation between the choice of a vegetarian or a vegan diet as a criterion of sustainability and the aspect of heartfulness. The following hypotheses will be investigated in detail:Because the choice of a vegetarian or vegan diet is related to environmental concerns and sustainable attitudes, which in turn are related to some aspects of heartfulness, we assume that vegetarians and vegans show higher scores in heartfulness than omnivores. Furthermore, it will be investigated whether there is a difference between vegans and vegetarians. Because veganism is a more sustainable behavior, we assume that the aspects of heartfulness are more pronounced in vegans than in vegetarians. For heartfulness, we chose the aspects of self-compassion, gratitude, compassion, and equanimity.Because the concept of heartfulness needs further investigation, we will explore the connection between heartfulness and the measured variables of the participants. We will examine if each aspect of heartfulness can be predicted by demographic variables (sex, age, education, and net income), diet-related variables (importance of nutrition, diet choice, and reason for diet choice), and the practice of mindfulness (meditation and movement-based meditation forms).

## 2. Method

### 2.1. Participants

The required sample size was calculated before data collection (see Appendix A). To achieve the targeted power for all analyses, we aimed for 10% more participants, resulting in 276 participants (92 participants per group). Overall, 419 persons participated. An additional 79 people had to be excluded for having answered less than 50% of the questions regarding one or multiple aspects of heartfulness. All demographic information is given in Appendix A. The participants had to be at least 18 years old to fill in the questionnaire. There were no other exclusion criteria.

### 2.2. Procedure

All questionnaires were implemented in SoSci Survey. The link was advertised to the participants via newsletter and social media. First, participants gave informed consent and then provided demographic information. After this, they completed the gratitude questionnaire, the self-compassion scale, the compassion scale, and the equanimity scale. Afterward, they were thanked for their participation. For each diet, there was a separate link. 

The study was conducted according to the ethical guidelines of the Helsinki Declaration and was approved by the ethics research board of the university (No. 22-3059-101). The study was preregistered at OSF. 

### 2.3. Measurements

#### 2.3.1. Demographic Questionnaire

Questions regarding sex, age, educational status, income, frequency of active meditation (never, once, xx minutes per month, or xx minutes per day) and yoga (never, once, xx minutes per month, or xx minutes per day), and the importance of nutrition (on a 5-point Likert scale ranging from (1) = *not important at all* to (5) = *very important*) were asked. Moreover, eating habits (vegan, vegetarian, or omnivorous) were reported. The participants were also asked to indicate their consideration of the environmental, ethical, and health-related reasons for their diet choice on a 5-point Likert scale ranging from (1) *= not relevant* at all to (5) *= very relevant*.

#### 2.3.2. Gratitude Questionnaire (GQ-5-G [51])

The original gratitude questionnaire, GQ-6, includes six items. Cronbach’s alpha was found to be 0.82 [52]. In this study, the German 5-item version, GQ-5-G, was used [51] to measure *gratitude*, because the validation study by Hudecek et al. [53] demonstrated that the model fit of GQ-6 was significantly improved once one item was eliminated. Participants had to rate each item on a 7-point Likert scale ranging from (1) = *strongly disagree* to (7) = *strongly agree.* A mean was calculated for the five responses.

#### 2.3.3. Self-Compassion Scale (SCS [41])

The German version, SCS-D, [54] was used for registering *self-compassion*. On the one hand, the SCS contains the positive components of *self-kindness*, *common humanity*, and *mindfulness*. On the other hand, it comprises the negative aspects of *self-judgment*, *isolation,* and *over-identification*. Answers were given on a Likert scale from (1) = *almost never to* (5) = *almost always*. A higher value on the negative scale indicates a higher level of *self-compassion*. Thus, the negative items were reverse coded for the analysis. The means of the positive and negative scales were separately used in the analysis, as recommended by Coroiu et al. [55].

#### 2.3.4. Compassion Scale [56]

The compassion scale (CS) is based on Neff’s theoretical model of self-compassion (2003). Compassion can be measured with the four subscales of *experiencing kindness*, *a sense of common humanity*, *mindfulness*, and (inverted) *disengagement, separation, and indifference toward the suffering of others*. The distinction between the compassion-subscale of *mindfulness* and FFMQ as standard measures of mindfulness ought to be noted as the items of the former all concerning interpersonal relationships, which none of the latter’s items do. The compassion scale consists of 16 items, with 4 items per subscale. Cronbach’s alpha and test–retest analyses were good for the overall compassion score, ranging from 0.78 to 0.90 across samples [56]. The English version was translated into German in the study by Siebertz et al. [35]. It was checked by all authors. In the study by Siebertz et al. [35], the omega of the total score was ω = 0.79, 95% CI = [0.73, 0.84], suggesting sufficient reliability. Participants had to answer how often they felt or behaved in the stated manner on a Likert scale from (1) = *almost never to* (5) = *almost always*. The mean of each compassion subscale was calculated.

#### 2.3.5. Equanimity Scale [57]

Juneau et al. [57] developed the two-factor equanimity scale with an even-minded state of mind (E-MSM) component and a hedonic independence component (HI). The E-MSM scale includes eight items, and the HI scale consists of six reversed items. Answers were given on a Likert-scale from (1) = *almost never to* (5) = *almost always.* The two components of equanimity displayed adequate internal consistency. For both scales, the mean was calculated. 

### 2.4. Statistical Analysis

Because some of the six dependent variables were correlated, three MANOVAs, each including two dependent variables, were calculated for the three diet choices, i.e., vegan, vegetarian, and omnivorous. One MANOVA included gratitude and compassion, one was calculated for the two equanimity scales and one analysed the differences in diet choice based on the two self-compassion scales. For the second hypothesis, six regression analyses were conducted with the nine predictors sex, age, education, and net income (demographic variables), importance of nutrition, diet choice, and reasons for diet choice (diet-related variables) and meditation and movement-based meditation forms (variables related to the practice of mindfulness) for each dependent variable.

## 3. Results

### 3.1. Demographic Data and Correlations

The correlations between the aspects of self-compassion, compassion, equanimity, and gratitude in vegans, vegetarians, and omnivores are presented in Appendix A. Because of the Bonferroni correction, only correlations with *p* < 0.003 should be interpreted. Vegans (Appendix A) showed significant correlations between both self-compassion scales on the one side and the equanimity even-minded state of mind scale on the other. In vegetarians, the equanimity hedonic independence scale was correlated with the equanimity even-minded state of mind (see Appendix A). Gratitude was correlated with the self-compassion negative scale and the equanimity even-minded state of mind scale. In omnivores, the self-compassion negative scale was correlated with compassion, while gratitude was correlated with compassion and the self-compassion positive scale (see Appendix A).

### 3.2. Differences in Age and Income Based on Diet Choice

We calculated an exploratory MANOVA to determine if there were differences in age and income based on the subjects’ diet choices. Indeed, the multivariate analysis with age and income as dependent variables using Pillai trace showed a significant effect of the diet choice (*F*(4, 830) = 418.557, *p* ≤ 0.001, ηp² = 0.082). The post hoc Bonferroni test showed significant differences in age between vegans and vegetarians (*p* ≤ 0.001, *M*_Diff_ = 6.91, 95%-CI [3.73, 10.09]) and between vegans and omnivores (*p* ≤ 0.001, *M*_Diff_ = 8.36, 95%-CI [5.19, 11.52]), with vegans being significantly older than both vegetarians and omnivores. Furthermore, it showed that vegans had a significantly higher income than vegetarians (*p* ≤ 0.001, *M*_Diff_ = 0.74, 95%-CI [0.48, 1.00]) and omnivores (*p* ≤ 0.001, *M*_Diff_ = 0.90, 95%-CI [0.64, 1.16])

### 3.3. Aspects of Heartfulness in Vegans, Vegetarians, and Omnivores

Because of the correlations explained above, the following three MANOVAs were calculated: Using Pillai trace, there were no statistically significant differences in the aspects of gratitude and compassion based on the subjects’ diet choice (*F*(4, 832) = 2.000, *p* = 0.093, ηp² = 0.010). No statistically significant differences based on the subject’s diet choice were found between the two equanimity scales using Pillai trace (*F*(4, 832) = 1.341, *p* = 0.253, ηp² = 0.006).

The multivariate analysis of the two scales of self-compassion using Pillai trace showed a significant effect of the diet choice (*F*(4, 832) = 4.125, *p* = 0.003, ηp² = 0.019). The post hoc Bonferroni test showed significant differences in the self-compassion positive scale between vegans and vegetarians (*p* = 0.003, *M*_Diff_ = 0.249, 95%-CI [0.068, 0.430]) and between vegans and omnivores (*p* = 0.008, *M*_Diff_ = 0.227, 95%-CI [0.046, 0.407]), which also supports the first hypothesis. Vegans scored significantly higher on the self-compassion positive scale than vegetarians and omnivores. We also found significant differences on the self-compassion negative scale between vegetarians and vegans (*p* = 0.001, *M*_Diff_ = 0.331, 95%-CI [0.110, 0.551]), as well as between omnivores and vegans (*p* = 0.031, *M*_Diff_ = 0.236, 95%-CI [0.016, 0.456]). This means that vegans scored significantly lower in negative self-compassion than vegetarians and omnivores. The negative self-compassion scale represents a lack of self-compassion. As highly negative self-compassion adversely influences heartfulness, this result also supports our first hypothesis.

### 3.4. Demographic, Diet-Related, and Mindfulness-Related Predictors of Aspects of Heartfulness

Regarding the second hypothesis, regressions were calculated for positive and negative self-compassion, compassion, gratitude, equanimity even-minded state of mind, and equanimity hedonistic independence. Sex, age, educational status, income, choice of diet, importance of nutrition, diet choice due to environmental and ethical reasons, diet choice due to health reasons, active meditation experience, and mindful movement experience were included as possible predictors in the regression analysis.

Assumptions such as linearity, homoscedasticity, normality, and independence were met for all six regressions. Multicollinearity did not pose a problem, with a minimum tolerance statistic of 0.488 and a maximum VIF of 2.047.

This multiple regression showed that 15% (adjusted R² = 13%) of the variance of the self-compassion positive scale was explained (*F*(10, 407) = 7.272, *p* < 0.001), with significant predictors being age, income, importance of nutrition, and diet choice due to health reasons (see Appendix A).

As presented in Appendix A, 11% (adjusted R² = 8%) of the variance of the self-compassion negative scale was explained (*F*(10, 407) = 5.035, *p* < 0.001). The only significant predictor was age.

For compassion, 13% (adjusted R² = 11%) of the variance was explained (*F*(10, 407) = 6.082, *p* < 0.001; see Appendix A). Sex, diet choice due to environmental and ethical reasons, and importance of nutrition were identified as significant predictors.

The multiple regression for gratitude showed that 15% (adjusted R² = 13%) of the variance of gratitude was explained (*F*(10, 407) = 7.223, *p* < 0.001), with significant predictors being importance of nutrition, diet choice due to environmental and ethical reasons, and diet choice due to health reasons (see Appendix A).

The multiple regression of the equanimity even-minded state of mind scale was not significant (R² = 4%, adjusted R² = 2%, F(10, 407) = 1.671, *p* = 0.085). For equanimity hedonic independence, the multiple regression was also not significant (R² = 4%, adjusted R² = 2%, *F*(10, 407) = 1.754, *p* = 0.067). Hence, they could not be interpreted.

## 4. Discussion

### 4.1. Correlations between Choice of Diet and Aspects of Heartfulness

To our knowledge, this is the first study examining the relation between the choice of a planetary diet as a criterion of sustainability and the aspect of heartfulness. Regarding the aspects of heartfulness, a significant correlation was found between gratitude and compassion. In general, the effects of these two aspects of heartfulness do not appear to be attributed to the choice of diet. As we suggested, vegetarians showed higher values in this aspect of heartfulness and compared to omnivores, chose to eat more sustainably. Vegetarians offering higher scores in diet choice due to environmental and ethical reasons also seems to point at this concern for the suffering of others and possible motivation to help, as compassion is defined [46] in the introduction.

We also found a significant correlation between the two aspects of equanimity, even-minded state of mind and hedonic independence, which is supported by their adequate internal consistency [57]. Contrary to our first hypothesis, no significant effects of the subjects’ diet choice on the two equanimity scales could be found.

The most apparent unambiguous result supporting our first hypothesis is the significant effect of diet choice on both self-compassion scales. The two scales were strongly correlated. Vegans showed significantly higher scores on the self-compassion positive scale than vegetarians and omnivores. As being vegan is, in this context, the most sustainable dietary alternative [24], it seems logical for vegans to score the highest on the self-compassion positive scale, because it represents heartfulness toward oneself [57]. Vegans also scored significantly lower on the self-compassion negative scale than vegetarians and omnivores. As the self-compassion negative scale represents the lack of self-compassion and includes the items of self-judgment, isolation, and over-identification [57], it negatively contributes to self-compassion and thus heartfulness overall. Both findings, effects on the self-compassion positive and negative scales, consequently, support our first hypothesis.

### 4.2. Differences in Age and Income Based on Diet

Because the frequency of a vegan diet in the general German population is lower than that of a vegetarian or omnivore diet [58], we had to use different social media platforms to find enough vegan participants. Consequently, we were forced to search outside the typical age range of college students. For that reason, the vegan individuals were significantly older than vegetarians and omnivores and, accordingly, had a significantly higher income [59]. These circumstances could potentially have influenced the results, as the biggest differences in the aspects of heartfulness were found between vegans and the other two diet choices. This occurrence does not comply with our hypothesis, as we expected the biggest difference to exist between the two planetary diet choices and omnivores. This indicates that the possible differences between the groups depended more on the platforms used and other characteristics of the participants than the diet groups. Another possible explanation for this result might be that the step from a vegetarian to a vegan diet is bigger than that of an omnivorous diet toward a vegetarian diet.

### 4.3. Variables Predicting the Concept of Heartfulness

To further understand heartfulness as a concept, we investigated the predictors of the six aspects of heartfulness. The positive predictors of the self-compassion positive scale were age, importance of nutrition, and diet choice due to health reasons, which means that the older the subject was, the higher they valued nutrition; the higher the importance of health reasons in their choice of diet was, the higher they tendentially scored in the self-compassion positive scale. It was shown that self-compassion and awareness increase with age [60]. In addition, it is understandable that those more focused on health reasons are more aware of themselves, care more for their own well-being [61], and thus show higher scores on the three facets of the self-compassion positive scale. Self-compassion was previously associated with health-promoting behaviors [57].

Furthermore, the lower the subject’s income was, the higher their score on the self-compassion positive scale was. One possible explanation for why a lower income predicted higher self-compassion might be that some people still associate self-compassion with being lazy or unproductive [57]. Following that thought process, one might drive and exhaust themself professionally and consequently have a higher income without considering a possibly improved well-being by practicing self-compassion. Those who give themselves that very kindness might not overwork themselves as much and thus have a lower income. Hereby, self-compassion functions as a protective factor. This application of self-compassion was shown to have significant effects in a recent study [62]. The lower the age is, the higher the negative self-compassion is. This is compatible with the previously mentioned result that self-compassion increases with age [60], as the negative self-compassion scale represents the lack of self-compassion. In consequence, the younger a person is, the stronger their negative self-compassion is.

Sex, diet choice due to environmental and ethical reasons, and importance of nutrition were significant predictors of the criterion of compassion. The positive predictors were importance of nutrition and diet choice due to environmental and ethical reasons. We defined compassion as concern for the suffering of another, which might result in the motivation to help [46]. Those who are more affected by the suffering of others are more focused on the environment and ethical aspects concerning their choice of diet. Sustainable behaviors might benefit the quality of life of others in the long term [63], which encompasses the key aspect of compassion, i.e., to help those suffering. In addition, those who value nutrition more highly show higher scores in compassion. The variable that negatively predicted compassion was sex, which means that women tended to be more compassionate than men. This finding was previously suggested by Neff and Pommier [57].

For the criterion of gratitude, the significant predictors were importance of nutrition, diet choice due to environmental and ethical reasons, and diet choice due to health reasons. All three were positive predictors, meaning that the higher the subjects’ values for these scales were, the higher their gratitude was. As environmental and ethical motives encompass a pursuit of sustainability [64], this result supports the finding mentioned above that feeling grateful directly influences sustainability and the selection of sustainable products [43]. There appears to be a relation between gratitude as an aspect of heartfulness and sustainable behavior. Furthermore, it is interesting that focusing on health reasons when choosing one’s diet predicts higher scores in gratitude. To feel grateful, a person must first recognize a positive outcome for oneself [40], which requires a certain awareness of one’s well-being and health. As gratitude is defined as an “empathic emotion” whose roots lie in the ability to empathize with others [41], a general awareness of one’s own body, sensations, thoughts, and emotions might be the base for relating to others. This finding is in line with a recent study investigating a heartfulness approach to reduce loneliness in high schoolers. It was demonstrated that by improving the connection to the self and by building social–emotional skills, loneliness scores in teenagers could be significantly lowered [65].

For both equanimity scales, the regressions were not significant, indicating that predictors other than demographic variables, diet-related variables, or the practice of mindfulness should be used to further investigate the concept of equanimity.

Notably, the importance of nutrition was a significant, positive predictor of the self-compassion positive scale, gratitude, and compassion. As aforementioned, mindfulness is defined as “paying attention in a particular way: on purpose, in the present moment, and nonjudgmentally” [41]. This awareness can manifest as an open and accepting attitude toward one’s own experiences [66]. Therefore, paying close attention to one’s nutrition and associated bodily sensations, as well as one’s health, is a form of mindfulness [67]. Because heartfulness represents the gentle emotional quality [36] of mindfulness, the association between increased mindful eating [68] and higher scores in positive self-compassion, gratitude, and compassion as aspects of heartfulness are logical. Higher self-compassion was previously connected with more mindful eating [69]. While the extent of mindful eating itself was not measured in this study, it has previously been shown that the importance one places on nutrition is related to mindful eating [70].

In general, demographic or diet-related variables proved to be better predictors of the aspects of heartfulness than mindfulness practice. The reasons for the subjects’ diet choices and the importance of their nutrition mainly seemed to predict their heartfulness. As previously mentioned, heartfulness is defined as the gentle emotional quality [36] of mindfulness, and it represents the caring, appreciative and nurturing attitude with which daily life is approached [37]. It seems that either this attitude encompasses a particular awareness of the relevance of ecological, ethical, and health-related standards or those standards encourage a more heartful approach to life. This awareness of both positive and negative aspects is also a known component of mindfulness [29] and is not only related to one’s own health and well-being but also to interpersonal contact [71] and the environment [72].

Altogether, the variance in heartfulness explained by our predictors was relatively small. This suggests that heartfulness is relatively stable concerning variations in the predictors. Most notably, it seems that mindfulness practice is not meaningfully related to heartfulness, and thus, mindfulness interventions might not be an appropriate way to promote heartfulness.

However, an explanation as to why mindfulness practice is not a significant predictor of heartfulness might be the way in which this aspect was registered. As the subjects were asked to indicate their mindfulness practice in “minutes per week” or “minutes per month”, there was no predetermined framework within which these values should lie. Consequently, the range and thus the variance of these values were extensive. This type of registration might have influenced the results and should be further investigated.

### 4.4. Limitations and Future Research

The paper’s focus was to investigate the relation between the choice of a planetary diet as a criterion of sustainability and the aspect of heartfulness, as well as the variables that predict the different aspects of heartfulness.

No causal conclusions could be drawn here. As only correlative relations were investigated, it is speculative whether the choice of diet affects heartfulness, the diet is affected by the aspects of heartfulness, or both. Furthermore, the method for recruiting for this study using social media and the university newsletter might have led to bias and statistical errors.

As there is no ultimate definition of the concept of heartfulness at this time, it might be reasonable to investigate either different aspects of the concept or other surveys for these aspects in future projects. In addition, the respective reason for the subjects’ diet choices could be more detailed in subsequent studies, and religious reasons for the diet might also be worth considering. It might be logical to remove the data of those subjects who are not free to choose their diet because of health restrictions.

Regarding the predictors of the aspects of heartfulness, different demographic or dietary variables should be researched in the future to achieve a better, more tangible, and more universal understanding of the concept of heartfulness. Heartfulness as an aspect of mindfulness and its relation to sustainability should be further investigated, not least because of their growing societal importance.

## 5. Conclusions

Our results show that vegans and vegetarians indicated higher scores than omnivores in some aspects of heartfulness, but only in some aspects, such as self-compassion. Furthermore, most aspects of heartfulness could either be predicted by demographic or diet-related variables. This study is the first one relating the aspects of heartfulness and choosing a plant-based diet. Further studies must follow to investigate this relation in more depth.

## Data Availability

Data and material are stored on OSF (https://doi.org/10.17605/OSF.IO/4GWCJ; https://doi.org/10.17605/OSF.IO/6PSWD).

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
