# Peer review of "Heartfulness in Vegans, Vegetarians, and Omnivores"

_ijerph, 2023, doi:10.3390/ijerph20064943_

Round 1

Reviewer 1 Report

Comments

Title: Heartfulness in vegans, vegetarians, and omnivores

While this article presents an interesting perspective, it could benefit from certain modifications to enhance its readability and cohesiveness for the intended audience.

Supplementary

1.          The variable numbers 2, 3... in Table S2a., Table S2b., Table S2c. The meaning of the numbers should be placed in the remarks of the table.

Abstract

2.          The tense in the abstract is inconsistent and it is recommended to seek assistance from a native English speaker to edit the entire article in order to improve readability and accuracy of the content.

Introduction

3.          Would it be possible to place the latest data on line 51? Additionally, I was wondering if there are any available data points from after 2020.

4.          References should be added for lines 88-89.

5.          The sentence is vague and needs to be restated "This attitude includes an inviting, curious factor but still a non-reactive, detached orientation.”

6.          Is there any relevant literature supporting lines 93-95 (Experimentally, no direct link between mindfulness and sustain...)?

7.          Please add a reference to this paragraph. “Jansen et al. discovered that attitudes towards sustainability were indirectly linked to aspects of mindfulness such as inner and outer awareness and insight, mediated by pro-social behavior.”

8.          Please add a reference to this paragraph. “…but already in 2004, Kabat-Zinn stated that mindfulness also has a gentle emotional…”

9.          Why was mindfulness mentioned in lines 116-117 and placed in this paragraph instead of the previous one? Its sudden introduction in this paragraph may disrupt the flow of the text for the reader.

10.      The two points of research hypotheses need to be concise and not include references to other people's literature.

11.      What is the relationship between mindfulness, dietary patterns, heartfulness, and nutrition, and a research framework is needed to explore this.

12.      In the introduction, I did not see whether the basic characteristics (sex, age, education, net income) of the participants in the past literature will affect " heartfulness ". It should describe what kind of situation the past literature has talked about, so your research hypothesis 2 is generated.

Methods

13.      Are underage questionnaires also included? There are no exclusion criteria, which is confusing.

14.      The 2.3. Procedure part should be moved to the back of 2.1. Participants, so that it can be read more smoothly.

Results

15.      The numbers inside R2 should be superscripts, and there are some minor errors in the content that require a complete re-check.

16.      The sentences in lines 317-319 are unclear. Please rephrase them.

17.      How do you know that these participants have adhered to the practice of mindful eating?

Discussion

18.      Some passages have problematic tenses, e.g. "we suggested, vegetarians show higher values in this aspect of heartfulness and, compared to omnivores, have made a choice to eat more sustainably."

19.      This sentence is unclear "This indicates our first hypothesis because, as for the self-compassion positive scale, vegans show higher scores in heartfulness than omnivores.

20.      The choice of diet may also be related to religious/beliefs. For example, Buddhists will also become vegetarians because they are worried about killing animals, so it may also affect the outcome, but this article does not mention it? What was the original consideration?

Reference

21.      Reference list needs to be updated, as most of the literature is outdated.

Author Response

Reply to Review 1

Title: Heartfulness in vegans, vegetarians, and omnivores

While this article presents an interesting perspective, it could benefit from certain modifications to enhance its readability and cohesiveness for the intended audience.

Supplementary

  1. The variable numbers 2, 3... in Table S2a., Table S2b., Table S2c. The meaning of the numbers should be placed in the remarks of the table.

Thank you for this comment. We have added the explanations for these numbers in the remarks of the table.

Abstract

  1. The tense in the abstract is inconsistent and it is recommended to seek assistance from a native English speaker to edit the entire article in order to improve readability and accuracy of the content.

We have changed the tense in the abstract. (line 15-31)

The language was revised again with the help of an English expert trained in psychology research. Some sentences were changed, rephrased, or condensed.

Introduction

  1. Would it be possible to place the latest data on line 51? Additionally, I was wondering if there are any available data points from after 2020.

Thank you for this comment. We have added the most recent data available regarding Germany, which is from 2022. (line 65-70)

  1. References should be added for lines 88-89.

We have added references for those two sentences, which are now in line 153-156.

  1. The sentence is vague and needs to be restated "This attitude includes an inviting, curious factor but still a non-reactive, detached orientation.”

We have changed the wording of this sentence to make it more unambiguous. (line 162-163)

  1. Is there any relevant literature supporting lines 93-95 (Experimentally, no direct link between mindfulness and sustain...)?

Thank you for this comment. We have added a reference for these two lines. (line 165-167)

  1. Please add a reference to this paragraph. “Jansen et al. discovered that attitudes towards sustainability were indirectly linked to aspects of mindfulness such as inner and outer awareness and insight, mediated by pro-social behavior.”

We have added a reference for this sentence. (line 169-173)

  1. Please add a reference to this paragraph. “…but already in 2004, Kabat-Zinn stated that mindfulness also has a gentle emotional…”

We have added the reference to this paragraph. (line 180-183)

  1. Why was mindfulness mentioned in lines 116-117 and placed in this paragraph instead of the previous one? Its sudden introduction in this paragraph may disrupt the flow of the text for the reader.

As mindfulness in the context of heartfulness reflects a slightly different aspect from mindfulness in general, we wanted to explain this interpretation of mindfulness in addition to the “universal” definition in the previous paragraph. If the disruption is too distracting, we could exclude the subject of mindfulness from this paragraph in a future version. (line 193-195)

  1. The two points of research hypotheses need to be concise and not include references to other people's literature.

We have cut all citations mentioned in the hypotheses paragraph and we have abbreviated this part in general.(line 273-289)

  1. What is the relationship between mindfulness, dietary patterns, heartfulness, and nutrition, and a research framework is needed to explore this.

We have tried to better explain the research gap and the research framework by adding some new references and demonstrating the relation between heartfulness, mindfulness, and sustainable diets. (line 255-269)

  1. In the introduction, I did not see whether the basic characteristics (sex, age, education, net income) of the participants in the past literature will affect " heartfulness ". It should describe what kind of situation the past literature has talked about, so your research hypothesis 2 is generated.

As there is little previous research regarding heartfulness we have taken an exploratory approach to this subject. For that reason, we could only add little literature for this hypothesis. (line 255-264)

Methods

  1. Are underage questionnaires also included? There are no exclusion criteria, which is confusing.

Underage questionnaires were not included. We have added this exclusion criteria as well as the fact that all those participants who did not answer at least 50% of questions for each aspect of heartfulness were excluded. (line 441-444)

  1. The 2.3. Procedure part should be moved to the back of 2.1. Participants, so that it can be read more smoothly.

We have changed the order of these two paragraphs. The procedure now comes before the questionnaires. The paragraph 2.2 Procedure is now in line 446-455 preceding the paragraph 2.3 Measurements starting in line 457.

Results

  1. The numbers inside R2 should be superscripts, and there are some minor errors in the content that require a complete re-check.

Thank you very much for this observation. We have changed the number inside R2 into a superscript. (lines 619-635) Unfortunately, we were not sure, what minor errors you mean. It would be great if you could further elaborate on these errors as we really want to change them.

  1. The sentences in lines 317-319 are unclear. Please rephrase them.

We have rephrased these sentences. (line 629-632)

  1. How do you know that these participants have adhered to the practice of mindful eating?

We have explained why we mentioned the practice of mindful eating by adding a reference to this paragraph. (line 782-809) Mindful eating itself was not necessarily relevant for our study, that’s why we didn’t measure it in the questionnaires, and it is not analyzed in the results. However, we mention mindful eating in the discussion in line 779-809 because it is used as a demonstration of the connection between higher self-compassion and the importance placed on nutrition by the participants. If you prefer, we could go into more detail on mindful eating.

Discussion

  1. Some passages have problematic tenses, e.g. "we suggested, vegetarians show higher values in this aspect of heartfulness and, compared to omnivores, have made a choice to eat more sustainably."

We have adjusted the tenses in the discussion and changed them to present tense for verbs regarding general results of the study.

  1. This sentence is unclear "This indicates our first hypothesis because, as for the self-compassion positive scale, vegans show higher scores in heartfulness than omnivores.

We have changed the phrasing of this sentence. (line 681-683)

  1. The choice of diet may also be related to religious/beliefs. For example, Buddhists will also become vegetarians because they are worried about killing animals, so it may also affect the outcome, but this article does not mention it? What was the original consideration?

This is true. Thank you for this advice. We will consider this point in further studies and added it to the limitation paragraph. (line 846-847)

We also explained why we chose to exclude possible religious beliefs from our study in the Introduction. (line 102-104)

Reference 

  1. Reference list needs to be updated, as most of the literature is outdated.

We have added some more recent literature. These are the new references: 16. Gwiozdzik et al. (2022), 17. Blaurock et al. (2021), 19. BMEL Ernährungsreport (2022), 34. Krupa-Kotara et al. (2023), 49. Desai et al. (2021), 63. Samrock et al. (2021), 66. Iyer et al. (2023).

Thank you for this advice.

Reviewer 2 Report

Dear Authors,

I was pleased to read your study that was submitted to me for review, it is an important and valuable item, however, it is not free of some errors that need to be completed or corrected:

Keywords need to be supplemented.

Wouldn't it be better to use the nomenclature traditional (customary) diet, since "omnivorous diet" does not function in the literature. 

In paragraph 1.3, it would be useful to include a recent 2023 study published in IJERPH, which found that women on a vegetarian diet feel less pressure to look good, which may be related to the fact that they are living a lifestyle that is now actively promoted on social media. And a 2022 study published in Nutrients, which addresses exposure to eating disorders depending on the diet used; traditional, vegetarian or medicinal. The most promising results came from a study of women precisely on a vegetarian diet, who were least likely to eat uncontrollably and emotionally, and were also least likely to suffer from diet-related diseases. Vegetarians most often choose to eat this way precisely for moral and ethical reasons - so it can be assumed that this is a conscious and sensitive group of consumers.

In addition, try to better demonstrate the research gap and provide a rationale for undertaking your research. 

Hypotheses and objectives are quite long-winded, perhaps it would be possible to describe it differently, the current way of writing makes perception of reception difficult.

The survey procedure should come before the description of the standardized questionnaires, because before I got to it the question arose "In the material describe how you recruited participants, whether it was a direct survey or the CAWI method, how the questionnaires were distributed," which I only got an answer to at the end of the chapter. 

Why was there no exclusion criteria provided? I think an age of at least 18 would have been reasonable, as there is a kind of vogue for vegetarianism in younger age groups. 

What was the participation in the study and were there any incorrectly completed questionnaires that you had to reject. 

In the introduction or in the material, better characterize German vegetarians. As a country with great cultural diversity, religious or cultural factors may have been crucial to the results. 

Why are the tables not in the main text only in the supplementary material, the lite text itself and the need to refer makes it difficult to read. 

The discussion was written correctly with reference to the relevant literature, which should, however, be updated with recent research on this study group. 

The method of recruiting for the study raises some concerns, may lead to bias and statistical error, this should be demonstrated in the limitations.   

The conclusions should be rewritten. 

Nevertheless, I congratulate the authors on a significant study. 

Regards

Author Response

Dear Authors,

I was pleased to read your study that was submitted to me for review, it is an important and valuable item, however, it is not free of some errors that need to be completed or corrected:

Thank you for this comment. We elaborate on each point in the following.

Keywords need to be supplemented.

We added two keywords. (line 33)

Wouldn't it be better to use the nomenclature traditional (customary) diet, since "omnivorous diet" does not function in the literature. 

Thank you for this valuable comment. We tried changing the nomenclature to traditional or customary but as there would be no corresponding noun (like “omnivore”) it would severely diminish the paper’s readability. For this, we decided to use the word “omnivore” instead. However, we are glad to receive further suggestions.

In paragraph 1.3, it would be useful to include a recent 2023 study published in IJERPH, which found that women on a vegetarian diet feel less pressure to look good, which may be related to the fact that they are living a lifestyle that is now actively promoted on social media. And a 2022 study published in Nutrients, which addresses exposure to eating disorders depending on the diet used; traditional, vegetarian or medicinal. The most promising results came from a study of women precisely on a vegetarian diet, who were least likely to eat uncontrollably and emotionally, and were also least likely to suffer from diet-related diseases. Vegetarians most often choose to eat this way precisely for moral and ethical reasons - so it can be assumed that this is a conscious and sensitive group of consumers.

Thank you, for this comment. We added the first study by Krupa-Kotara et al. (2023) in paragraph 1.3 (line 173–178). The second study by Gwioździk et al. (2022) was added to paragraph 1.1 (line 88-93). Because of its reference to possible reasons for a vegetarian diet, we assumed it thematically belongs in this part of the introduction. However, we are willing to mention it in paragraph 1.3 also, should you prefer it there.

In addition, try to better demonstrate the research gap and provide a rationale for undertaking your research. 

We have supplemented additional explanations regarding the research gap. We have added some literature and tried to demonstrate the added value of our study, (line 259-269). If in your opinion this part is not detailed enough, we could also make it into its own paragraph.

Hypotheses and objectives are quite long-winded, perhaps it would be possible to describe it differently, the current way of writing makes perception of reception difficult.

We have cut several citations mentioned in the hypotheses paragraph and we have abbreviated this part in general. (line 273-289)

The survey procedure should come before the description of the standardized questionnaires, because before I got to it the question arose "In the material describe how you recruited participants, whether it was a direct survey or the CAWI method, how the questionnaires were distributed," which I only got an answer to at the end of the chapter. 

Thank you for this helpful comment. We have changed the order of these paragraphs; the procedure now comes before the questionnaires. The paragraph 2.2 Procedure is now in line 446-455 preceding the paragraph 2.3 Measurements starting in line 457.

Why was there no exclusion criteria provided? I think an age of at least 18 would have been reasonable, as there is a kind of vogue for vegetarianism in younger age groups. 

We supplemented the exclusion criteria. One had to be 18 years of age to participate in the questionnaire, which we added in line 443.

What was the participation in the study and were there any incorrectly completed questionnaires that you had to reject. 

We supplemented the number of people we had to exclude due to incorrectly filling in the questionnaire. One had to answer more than 50% of questions for each aspect of heartfulness to avoid being excluded, which was also added in this paragraph, (line 441-442)

In the introduction or in the material, better characterize German vegetarians. As a country with great cultural diversity, religious or cultural factors may have been crucial to the results. 

We added some literature regarding German vegetarians and mentioned some facts about German vegetarians and omnivores in particular, (line 94-104)

Why are the tables not in the main text only in the supplementary material, the lite text itself and the need to refer makes it difficult to read. 

Thank you for this comment. We initially didn’t place the tables in the main manuscript as to keep the word count as low as possible because it was already relatively high. If that is a problem, we are willing to add the tables to the main text immediately.

The discussion was written correctly with reference to the relevant literature, which should, however, be updated with recent research on this study group. 

Thank you very much for this valuable information. We added some more recent literature to the discussion. Three newly added more recent studies are mentioned in line 255-258 in the introduction and line 737-738 as well as line 766-769 in the discussion.

The method of recruiting for the study raises some concerns, may lead to bias and statistical error, this should be demonstrated in the limitations.   

Thank you for this comment. We added these concerns to the limitations section. (line 841-842)

The conclusions should be rewritten. 

We have rewritten the conclusions. (line 855-860)

Nevertheless, I congratulate the authors on a significant study. 

Thank you very much for this comment.

Round 2

Reviewer 1 Report

Thank you for taking the time to improve the article. Your efforts have resulted in a better and more polished piece of writing. Your contribution is greatly appreciated.

Reviewer 2 Report

Dear Editor,
The paper submitted to me for review, "Heartfulness in vegans, vegetarians, and omnivores" by Antonia Voll, Leonardo Jost, Petra Jansen, has been appropriately revised and corrected. So I recommend its acceptance for publication in IJERPH after final proof. 

Greetings